

# Comprehensive analysis of metastasis-related genes reveals a gene signature predicting the survival of colon cancer patients

Haotang Wei[1,*], Jilin Li[2,*], Minzhi Xie[2], Ronger Lei[3] and Bangli Hu[2]

[1] Department of Gastrointestinal Surgery, Third Affiliated Hospital of Guangxi Medical University, Nanning, China
[2] Department of Research, Affiliated Tumor Hospital of Guangxi Medical University, Nanning, China
[3] Department of Gastroenterology, First Affiliated Hospital of Guangxi Medical University, Nanning, China
* These authors contributed equally to this work.

Corresponding author
Bangli Hu, hubangli@gxmu.edu.cn

## ABSTRACT

**Objective:** The mechanism underlying colon cancer metastasis remain unclear. This study aimed to elucidate the genes alteration during the metastasis of colon cancer and identify genes that crucial to the metastasis and survival of colon cancer patients.

**Methods:** The dataset of primary and metastasis tissue of colon cancer, and dataset of high and low metastasis capability of colon cancer cells were selected as training cohort, and the overlapped differentially expressed genes (DEGs) were screened from the training cohort. The functional enrichment analysis for the overlapped DEGs was performed. The prognostic value of overlapped DEGs were analyzed in The Cancer Genome Atlas dataset, and a gene signature was developed using genes that related to the overall survival (OS). The prognostic value of the gene signature was further confirmed in a validation cohort.

**Results:** A total of 184 overlapped DEGs were screened from the training cohort. Functional enrichment analysis revealed the significant gene functions and pathways of the overlapped DEGs. Four hub genes (3-oxoacid CoA-transferase 1, actinin alpha 4, interleukin 8, integrin subunit alpha 3) were identified using protein–protein network analysis. Six genes (aldehyde dehydrogenase 2, neural precursor cell expressed, developmentally down-regulated 9, filamin A, lamin B receptor, twinfilin actin binding protein 1, serine and arginine rich splicing factor 1) were closely related to the OS of colon cancer patients. A gene signature was developed using these six genes based on their risk score, and the validation cohort indicated that the prognostic value of this gene signature was high in the prediction of colon cancer patients.

**Conclusions:** Our study demonstrates a gene profiles related to the metastasis of colon cancer, and identify a six-gene signature that acts as an independent biomarker on the prognosis of colon cancer.

## INTRODUCTION

Colorectal cancer (CRC) is one of the leading malignant cancers in the world, and colon cancer accounts for a large part of CRC (*Siegel et al., 2017*; *Siegel, Miller & Jemal, 2017*). During the last three decades, with the development of new therapies, such as asbevacizumab and anti-epidermal growth factor receptor antibody (*Castro et al., 2013*; *Knijn, Tol & Punt, 2010*), survival outcomes for colon cancer patients with localized and regional disease have improved greatly. However, cancer metastasis remains one of the main causes of death in these patients. One studied showed that the prognosis of patients with distant metastasis remained poor compared to the prognosis of patients with early-stage colon cancer (*Siegel, Miller & Jemal, 2017*). Accumulation of genetic mutations is recognized as one of the important causes for the pathogenesis and progression of colon cancer (*Sameer, 2013*). Hence, exploring the mechanisms of cancer metastasis and searching suitable predictors are crucial to the diagnosis and treatment of colon cancer.

Previously, the tumor, node, metastasis (TNM) stage and pathological characteristics of colon cancer have been used to predict the prognosis and facilitate treatment for colon cancer patients. However, there are some limitations in using these methods to assess patients (*Marzouk & Schofield, 2011*). Recently, several novel biomarkers have been tested with the aim of improving the prediction of therapeutic response and prognosis of colon cancer patients (*Demirkol et al., 2017*; *Hu et al., 2014*; *Xu et al., 2017*). These have aided in the diagnosis and treatment of colon cancer, but the results were inconsistent and need to be further studied.

Thus far, metastasis is a major factor for poor prognoses of colon cancer patients, and the liver is the most common organ affected. However, the molecular mechanism underlying distant metastasis remains unclear. Therefore, a comprehensive analysis of the molecular alterations to identify prognostic indicators is key for the management of colon cancer patients with distant metastasis. In this study, we used the colon cancer data from gene expression omnibus (GEO) and The Cancer Genome Atlas (TCGA) and analyzed the data of patients with colon cancer using primary tumor samples and liver metastasis samples to unveil the genes key to the development colon cancer metastasis and the potential prognostic indicators.

## MATERIALS AND METHODS

### Patient datasets

The colon cancer tissue and cells microarray data (GSE40367 (*Roessler et al., 2015*) and GSE2509 (*Provenzani et al., 2006*)) were retrieved and downloaded from the GEO (http://www.ncbi.nlm.nih.gov/geo/) database of National Center for Biotechnology Information as the training cohort. The GSE40367 dataset includes seven colon adenocarcinoma (COAD) with liver metastasis species and eight COAD primary tumor species. The GSE2509 dataset includes two colon cancer cell lines (SW480: low metastasis capability and SW620: high metastasis capability). The prognostic value of genes was analyzed using the data of COAD from TCGA. To confirm the results from training cohort, we used the GSE41258 (*Sheffer et al., 2009*) dataset that includes 390 species

as a validation cohort. Because the data were download freely from GEO and TCGA database, approval of the ethics committee of Guangxi Medical University was not needed.

### Identification of overlapped DEGs

R language software (version 3.4.2) and Bioconductor packages were applied to screen the differentially expressed genes (DEGs) between primary tumor tissue and liver metastasis tissue in GSE40367. The DEGs between SW480 and SW620 in GSE2509 were also screened. Genes that fulfilled the criteria of having a $p$-value $< 0.05$ and $|logFC| \geq 1$ were defined as the DEGs. Then, the intersected DEGs of GSE40367 and GSE2509 were defined as overlapped DEGs. The probe level GSE data were converted into gene expression values to measure each gene before the screening of DEGs. If one gene corresponded to multiple probe sets, we used the average data of the multiple probes as the gene expression values (*Qin et al., 2012*). We also eliminated genes that had over >20% values of the total samples as a previous study did (*Liew, Law & Yan, 2011*). After pre-processing the data, $t$-test methods were used to screen the DEGs using a limma package.

### Functional enrichment analysis of overlapped DEGs

Gene ontology (GO) includes three categories, namely, the biological process (BP), molecular function (MF), and the cellular component (CC). To investigate the functional level of DEGs, the genes underwent GO analysis using the Database for Annotation, Visualization, and Integrated Discovery (DAVID, https://david.ncifcrf.gov/). The significant GO categories were defined as those with $p < 0.05$. Following this, Kyoto Encyclopedia of Genes and Genomes (KEGG) pathway analysis was conducted to identify significant pathways with gene enrichment using a DAVID online tool. We defined pathways as significant if the $p$-value was $<0.05$.

### Integration of PPI network and subnetwork analysis

Protein–protein interaction (PPI) networks was used to identity key genes and important gene modules which are involved in cancer development by assessing gene interaction levels. In this study, the data of PPI network for overlapped DEGs was acquired from the Search Tool for the Retrieval of Interacting Genes (STRING) database (http://www.string-db.org/). Then, Cytoscape software (version 3.5.2) was used for the establishment of a PPI network for all overlapped DEGs. After the establishment of a PPI network, module analysis was carried out by molecular complex detection to detect the gene modules of the PPI network. The hub of genes of each module was identified based on the score of each gene in the module.

### Acquisition of a gene signature from the training cohort

The association of overlapping DEGs with the overall survival OS of colon cancer patients was analyzed in the COAD dataset from the TCGA database using a Cox regression analysis in the survival package of R. The gene with $p < 0.05$ was considered to be an independent prognostic factor. To estimate the relative contribution of multiple genes

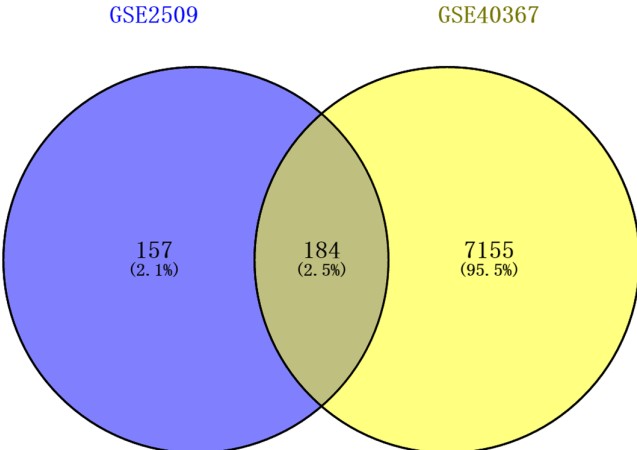

**Figure 1 Venn diagrams showing intersected genes overlapping between time-series genes (TSGs) and differentially expressed genes (DEGs).**

for survival prediction of colon cancer patients, the hub genes were applied to develop a prognostic model using the risk scoring system. In brief, the risk score system was calculated based on a linear combination of the expression level multiplied by the regression coefficient derived from the multivariate Cox regression model (the β value) with the following formula as previously reported: Risk score = expression of $Gene_1 \times \beta_1 Gene_1$ + expression of $Gene_2 \times \beta_2 Gene_2$ + ⋯ expression of $Gene_n \times \beta_n Gene_n$ (*Hur et al., 2015*). Using the median risk score as the cutoff, patients were divided into high-risk and low-risk groups. Kaplan–Meier curves were used to estimate the difference in survival for patients between the high-risk group and low-risk groups. $p < 0.05$ was defined as significantly different. Time-dependent receiver operating characteristic (ROC) curve analysis for the gene signature was performed using the R package "survivalROC" (*Heagerty, Lumley & Pepe, 2000*). All statistical analyses were performed using R software and Bioconductor.

# RESULTS

## Overlapped genes of CRC cells and tissues

The DEGs of GSE2509 and GSE40367 were screened on the basis of the selection criteria after preprocessing the raw data. A total of 341 DEGs between colon cancer cells lines SW420 and SW680 cells were identified in GSE2509 dataset, and 7339 DEGs between primary and metastasis tumor specimens were identified in the GSE40367 dataset. By overlapping the DEGs from the two datasets, we obtained 184 overlapped genes that differentially expressed in both CRC cells and tissues. The result is shown in Fig. 1.

## GO and KEGG enrichment analysis

The gene functions of the 184 overlapped DEGs were then analyzed by GO and KEGG enrichment analysis. Using the DAVID online tools, we found that the most enriched GO terms of DEGs related to BP were Signal transduction, and the MF was Protein
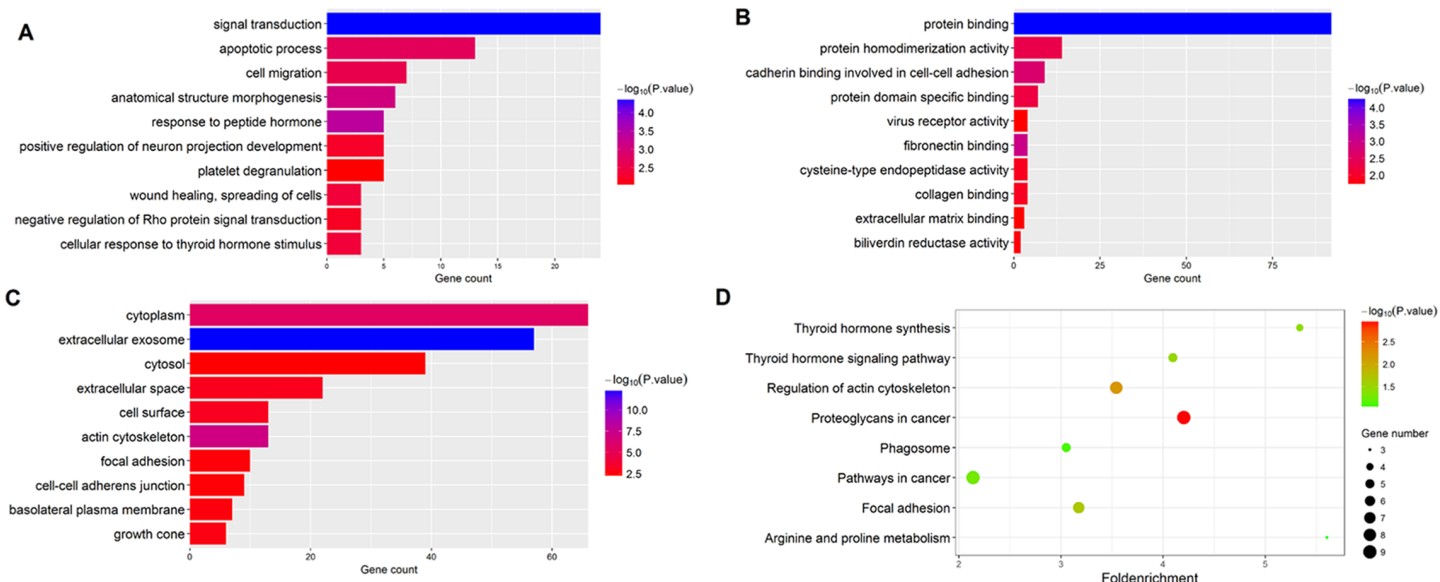

**Figure 2 Functional enrichment analysis of overlapped DEGs.** (A) GO enrichment analysis of overlapped DEGs in biological processes; (B) GO enrichment analysis of overlapped DEGs in molecular function; (C) GO enrichment analysis of overlapped DEGs in cellular component; (D) KEGG analysis of overlapped DEGs.

binding, and the CC was Cytoplasm. The KEGG pathway analysis based on the GO results revealed that Thyroid hormone synthesis was the most significant pathway of the overlapped DEGs. The results are shown in Fig. 2.

## PPI network and module screening analysis

Using data from the STRING database, a PPI network for the 184 DEGs consisting of 133 nodes and 138 edges was constructed using Cytoscape software. The overall PPI network is shown in Fig. 3A. Following this, the plug-in MCODE was used to detect the modules in the network, and four modules from the PPI network were identified, with the 3-oxoacid CoA-transferase 1 (*OXCT1*) (sore: 4), actinin alpha 4 (ACTN4) (sore: 3), interleukin 8 (*IL-8*) (sore: 3), integrin subunit alpha 3 (*ITGA3*) (sore: 3) as the hub genes of each module. The top three modules are shown in Figs. 3B–3D.

## Prognostic value of overlapped DEGs

The prognostic value of the 184 overlapping DEGs was analyzed using the COAD dataset of TCGA by multiple Cox regression analysis after adjusting the data for age, sex, and TNM stage. The results showed that only aldehyde dehydrogenase 2 (*ALDH2*), neural precursor cell expressed, developmentally down-regulated 9 (*NEDD9*), filamin A (*FLNA*), lamin B receptor (*LBR*), twinfilin actin binding protein 1 (*TWF1*), and serine and arginine rich splicing factor 1 (*SRSF1*) were independent genes that associated with the OS of colon cancer patients, with β values of −1.343, −0.051, 0.492, −0.020, −0.181, and −1.938, respectively. We developed a six-gene signature by calculating the risk score of each gene, and divided the patients into a high-risk group and a low-risk group on the basis of the median risk score (Fig. 4A), the survival status, and the genes expression level shown in Figs. 4B–4C.

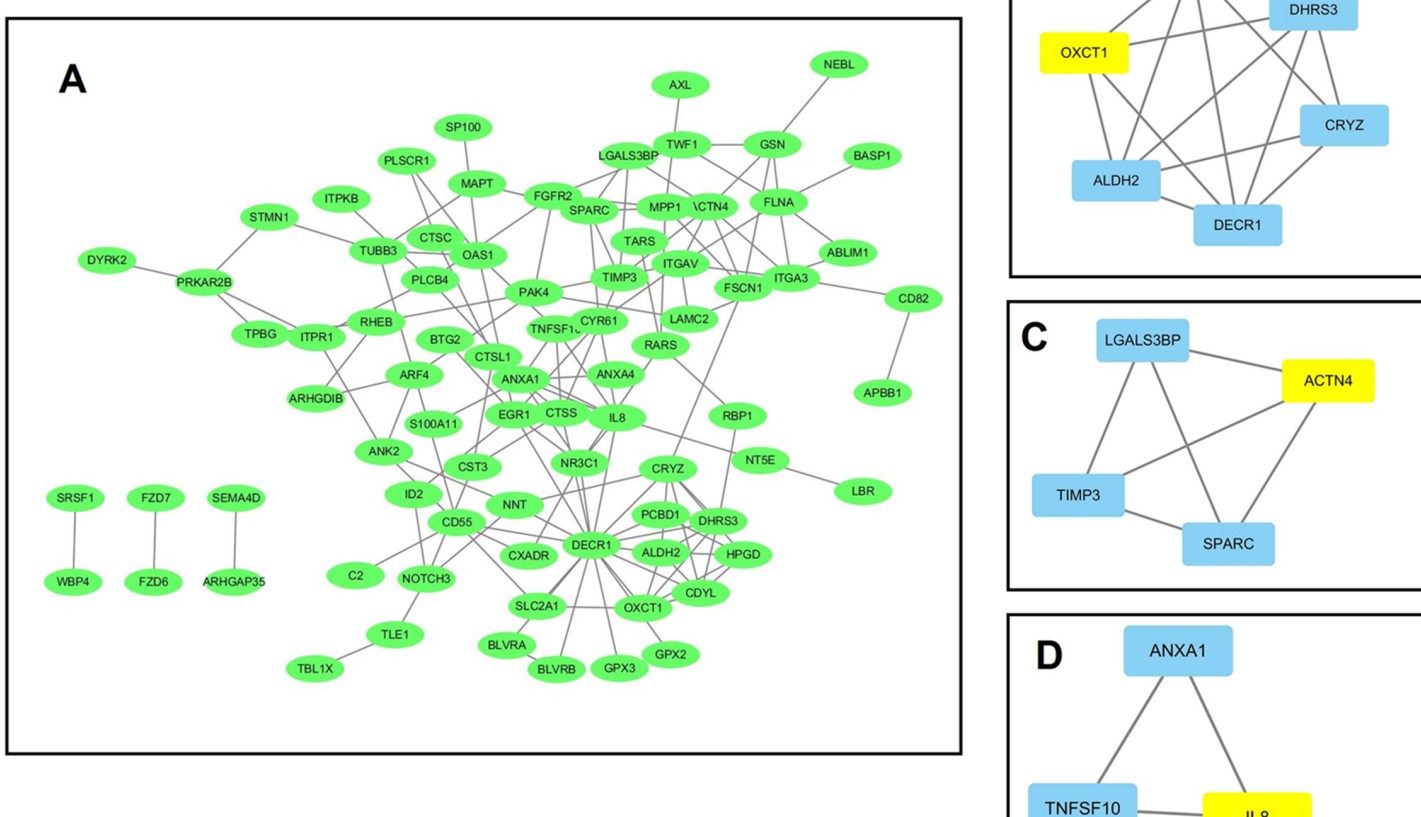

**Figure 3 The protein–protein interaction (PPI) network analysis of the 326 overlapped DEGs.** (A) PPI network for the overlapped DEGs; (B–D) three major submodules of the PPI network.

The survival analysis revealed that this six-gene signature within the high-risk group predicted poor OS of colon cancer patients compared to that in the low-risk group, with $p < 0.001$ (Fig. 5A). Using survival ROC analysis, we found that the risk score of this six-gene signature could moderately predict the 1-, 3-, 5-year OS of colon cancer patients, as the value of the area under ROC curve was 0.686, 634, and 618, respectively (Fig. 5B).

## Validation cohort confirms the prognostic value of the six-gene signature

The prognostic value of the six-gene signature for the OS of colon patients was further determined in the validation cohort (GSE41258 datasets, 390 colon patients, mean follow-up 65.3 months). By using the same risk score model and cutoff value derived from the training cohort, 390 patients of the validation cohort were classified into either a high-risk group ($n = 195$) or a low-risk group ($n = 195$). Consistent with results of the training cohort, the results of this six-gene signature indicated an obvious difference between the high-risk group and low-risk group with regard to the OS of colon cancer patients ($p = 0.005$, log-rank test).

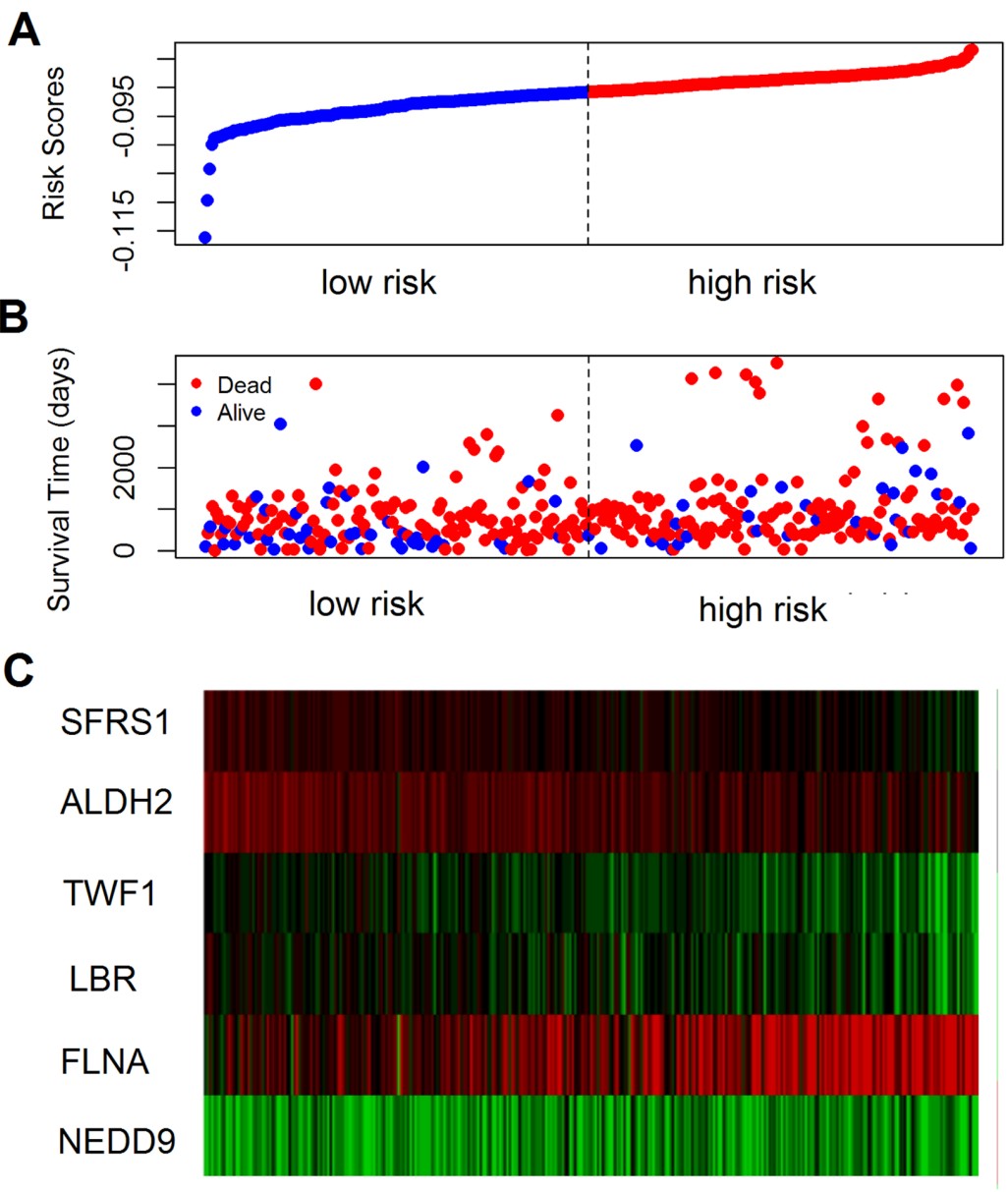

**Figure 4** **The prognostic performance of the six-gene signature of colon cancer.** (A) Risk score curve of the six-gene signature; (B) patient survival status and time distribution by risk score; (C) heatmap showed the expression of six genes by risk score.

## DISCUSSION

Metastasis accounts for 90% of the mortalities of colon cancer patients and thus has become the most lethal characteristic of colon cancer (*Li et al., 2017*). Colon cancer patients with localized and regional disease have a good prognosis (the 5-year survival rate is up to 91.1%), but patients with distant metastasis have a much worse prognosis (the 5-year survival rate has dropped to 13.3%) (*Siegel et al., 2017*). Furthermore, the failure of treatment is mostly caused by the metastatic dissemination of primary tumors (*Deliu, Georgescu & Bezna, 2014*; *Stein & Schlag, 2007*). At a molecular level, distinct

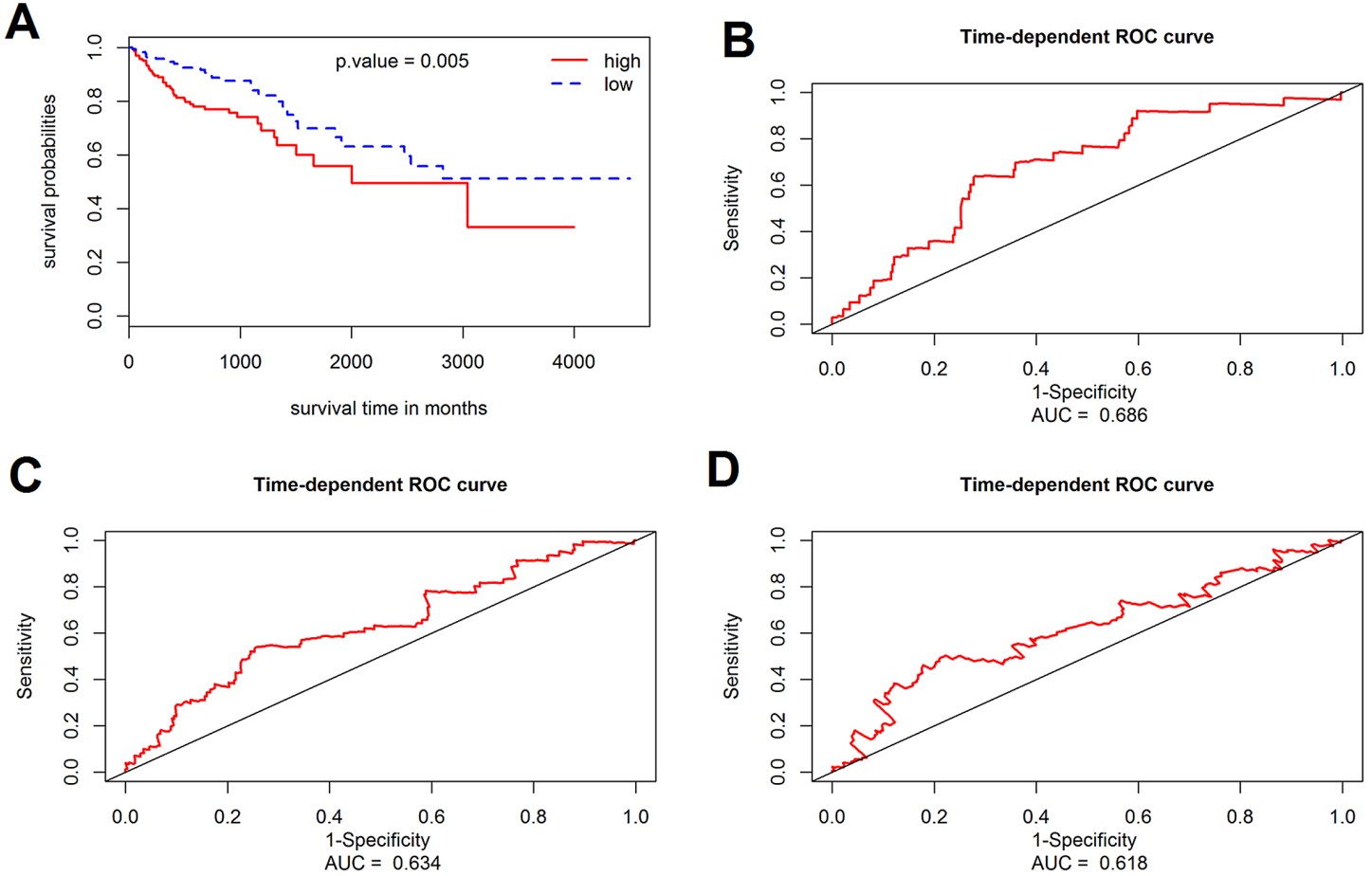

**Figure 5 Survival analysis for the gene signature.** (A) The Kaplan–Meier test of the risk score for the overall survival of colon cancer between high-risk and low-risk patients; (B–D) the prognostic value of the risk score showed by the time-dependent receiver operating characteristic (ROC) curve for predicting the 1, 3, 5 years overall survival, respectively.

metastasis of colon cancer is molecularly and clinically distinct from the primary site of origin (*Zarour et al., 2017*). Thus, analyzing the molecular alteration of colon cancer with distinct metastasis is beneficial for the identification of candidate targets of early diagnosis and treatment of advanced stage colon cancer.

To date, some biomarkers have been identified as candidate targets for early diagnosis and treatment of colon cancer and rectal cancer, including genes, miRNA, lncRNA, and the related signatures. Some gene signatures related to the metastasis potential of each tumor have been described, with promising results. *Vellinga et al. (2017)* designed a lymphangiogenic gene set, applied it to large datasets of CRC, and found that this lymphangiogenic gene set was related to a worse prognosis. *Rokavec et al. (2017)* reported a single gene, RNA binding motif protein 47, down-regulated during CRC progression may promote epithelial–mesenchymal transition and metastasis. Furthermore, proteomic studies used exosomes that from cancer cell lines and identified four candidate genes MET proto-oncogene, receptor tyrosine kinase, S100A8, S100A9, tenascin C associated with CRC metastasis (*Ji et al., 2013*). Other biomarkers, such as a four-miRNA signature

(let-7i, miR-10b, miR-221, and miR-320a) (*Hur et al., 2015*), and a six-lncRNA signature were reported to be promising biomarkers for the metastasis and prognosis of CRC (*Hu et al., 2014*).

In this study, we identified four hub genes from the subnetwork of the PPI network: OXCT1, ACTN4, IL-8, and ITGA3. OXCT1 is a key enzyme involved in the process of ketone body metabolism, and catalyzes the first and rate-determining step of ketolysis. OXCT1 can be converted into acetyl-CoA during the process of metabolism and takes part in the tricarboxylic acid cycle for the oxidation and the production of adenosine triphosphate (ATP) (*Zhang & Xie, 2017*). The role of OXCT1 has been implicated in several cancers, included CRC, and OXCT1 was overexpressed in the metastatic CRC cell line CC-M3 (*Lee et al., 2016*). ACTN4 is a non-muscle-type alpha-actinin, which plays an important role in regulating cytoskeleton organization and is involved in transcriptional regulation of gene expression. ACTN4 encodes a nonmuscle, alpha-actinin isoform that is concentrated in the cytoplasm; participates in metastatic processes; and facilitates the motility, invasion, and metastasis of cancer cells. In CRC, ACTN4 was reported to promote CRC cell line invasion by suppressing focal adhesion maturation (*Fukumoto et al., 2015*). ITGA3 belongs to the integrin family, joins a $\beta_1$ subunit to form an intact integrin and interacts with several extracellular matrix proteins (*Nagata et al., 2013*). ITGA3 was found to be associated with lymphatic dissemination and local invasiveness in cancers. One study showed that ITGA3 was overexpressed in stages III vs I of CRC patients and is related to OS and disease-free survival (*Linhares et al., 2015*). IL-8 is an important pro-inflammatory chemokine and plays a role in the recruitment of leukocytes to the sites of infection or tissue injury. Growing evidence suggests that paracrine signaling by tumor-derived IL-8 promotes the trafficking of neutrophils and myeloid-derived suppressor cells into the tumor microenvironment, which is associated with dampened anti-tumor immune responses (*David et al., 2016*). *Lambrechts et al. (2015)* observed that IL-8 plasma levels at baseline and subsequent increases in IL-8 were associated with worse progression-free survival in metastatic CRC patients. These studies confirmed our results that the hub genes of the PPI network were crucial to the metastasis of colon cancer, but further studies are warranted to determine the underlying mechanism.

Similar to the previous studies, in this study, a signature constructed by six genes was shown to be a good predictor of the OS of patients with colon cancer. Among these six genes, the ALDH2, NEDD9, SRSF1, and FLNA were reported to be associated with the CRC in several studies. ALDH2 is essential for the metabolism and detoxification of a wide range of endogenous and exogenous aldehyde substrates. ALDH2 is the rate-limiting enzyme in the ethanol metabolism, oxidizing acetaldehyde to acetic acid both in the liver and other tissues (*Chen, Joshi & Mochly-Rosen, 2016*). As a novel biological marker, ALDH2 is an attractive prospect in the screening, diagnosis, and evaluation of the prognosis of many diseases, and the genetic polymorphism of ALDH2 significantly correlated with the susceptibility to CRC (*Li et al., 2016*). NEDD9 is a non-catalytic scaffolding protein that assembles complexes involving oncogenic kinases and regulates the magnitude and duration of cell signaling cascades that control multiple processes important to the development

and progression of tumors (*Shagisultanova et al., 2015*). Studies have shown that downregulation of NEDD9 by apigenin can suppresses migration, invasion, and metastasis of CRC cells (*Dai et al., 2016*). With regard to SRSF1, one study reported that phosphorylation of SRSF1 regulated alternative splicing of tumor-related Rac1b in CRC cells (*Goncalves et al., 2014*). SRSF1 is classified as an exonic splicing enhancer and recognizes degenerate purine-rich sequence motifs. In addition, it can promote the recognition of both constitutive and alternative exons during the process of spliceosomal assembly (*Sanford et al., 2009*). FLNA is an actin-binding protein expressed ubiquitously in the body, involved in many cell-signaling pathways, and it is important in the maintenance of cell shape and motility. The mutation of FLNA has been shown to be associated with the neuronal migration, vascular function, connective tissue integrity, and skeletal development (*Shelmerdine et al., 2017*); FLNA showed low expression in CRC patients and was significantly correlated with the incidence and development of CRC (*Tian et al., 2015*). LBR, a transmembrane protein of the inner nuclear membrane, interacts with heterochromatin and B-type lamins through its nucleoplasmic amino-terminal domain and is phosphorylated throughout the cell cycle (*Duband-Goulet, Courvalin & Buendia, 1998*). TWF1 belongs to the ADF-H family and is a conserved actin-binding protein. TWF1 regulates diverse morphological processes through sequestering ADP-actin monomers or capping filament barbed ends (*Paavilainen et al., 2007*). However, no study has yet reported the role of LBR and TWF1 in CRC and therefore their role needs be investigated further in colon cancers.

Compared with previous studies, this study screened the metastasis-related genes by overlapping the DEGs from cancer tissues and cell lines. These tissues included primary colon cancer tissues and liver metastasis tissues, and the cell lines included primary (SW480) and metastatic (SW620) human isogenic CRC cell lines. Thus, the overlapped DEGs could more reliably reflect the gene alteration in metastatic colon cancer. Some limitations of this study need to be noted. First, although metastasis-related genes of colon cancer were identified and their prognostic value was validated in our study, the results were calculated from microarray or RNA-sequencing technique datasets and thus lack of functional validation of the target genes is one of the major limitations of this study. Therefore, thorough functional experiments for these genes and corresponding downstream events to reveal novel diagnostic and therapeutic targets for colon cancer is necessary. Second, the development of colon cancer metastasis can be caused by many factors, such as KRAS, BRAF mutation, and microsatellite instability, which have been proven to be closely related to colon cancer, but because of the limitations of the datasets, we did not perform a stratified analysis on the basis of these factors. Future studies should be conducted to explore the difference under different conditions. Third, the mean time of follow-up in the validation cohort was 65.3 months; thus, a study including a longer follow-up time is warranted to validate our results in the future. Fourth, this study only used a GSE dataset to perform bioinformatics to search for biomarkers. Future studies should implement Partek Genomics software and Illumina Correlation Engine for bioset analyses.

## CONCLUSIONS

In this study, we screened gene profiles involved in the metastasis of colon cancer and identified four hub genes from the profiles. We also identified and validated a six-gene signature that can serve as a prognostic indicator of colon cancer. Some genes that have not yet been proven to be associated with colon cancer metastasis may represent new therapeutic targets.

## ABBREVIATIONS

| | |
|---|---|
| **CRC** | colorectal cancer |
| **DEGs** | differentially expressed genes |
| **GO** | Gene ontology |
| **KEGG** | Kyoto Encyclopedia of Genes and Genomes |
| **OS** | overall survival |
| **ROC** | receiver operating characteristic curve |
| **OXCT1** | 3-oxoacid CoA-transferase 1 |
| **ACTN4** | actinin alpha 4 |
| **IL-8** | interleukin 8 |
| **ITGA3** | integrin subunit alpha 3 |
| **ALDH2** | aldehyde dehydrogenase 2 |
| **NEDD9** | neural precursor cell expressed, developmentally down-regulated 9 |
| **FLNA** | filamin A |
| **LBR** | lamin B receptor |
| **TWF1** | twinfilin actin binding protein 1 |
| **SRSF1** | serine and arginine rich splicing factor 1 |
| **TNC** | tenascin C |
| **RBM47** | RNA binding motif protein 47 |
| **MET** | MET proto-oncogene, receptor tyrosine kinase. |

### Funding

This work was supported by the Scientific research and technology development project of Nanning (20173018-3; 20163120). The funders had no role in study design, data collection and analysis, decision to publish, or preparation of the manuscript.

### Grant Disclosures

The following grant information was disclosed by the authors:
Scientific research and technology development project of Nanning: 20173018-3; 20163120.

### Competing Interests

The authors declare that they have no competing interests.

## Author Contributions

- Haotang Wei conceived and designed the experiments, performed the experiments, analyzed the data, contributed reagents/materials/analysis tools, prepared figures and/or tables, authored or reviewed drafts of the paper, approved the final draft.
- Jilin Li performed the experiments, analyzed the data, authored or reviewed drafts of the paper, approved the final draft.
- Minzhi Xie performed the experiments, analyzed the data, authored or reviewed drafts of the paper, approved the final draft.
- Ronger Lei conceived and designed the experiments, performed the experiments, analyzed the data, authored or reviewed drafts of the paper, approved the final draft.
- Bangli Hu conceived and designed the experiments, contributed reagents/materials/analysis tools, prepared figures and/or tables, authored or reviewed drafts of the paper, approved the final draft.

## Data Availability

The raw data can be downloaded from the GEO database: accession numbers: GSE40367 and GSE2509, and TCGA (colon cancer).

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
