# Peer review of "Comprehensive analysis of metastasis-related genes reveals a gene signature predicting the survival of colon cancer patients"

_PeerJ, doi:10.7717/peerj.5433_

## Round 0.1 · original submission · Major Revisions

This analysis is of merit. Please address the comments from reviewers point by point. Two reviewers recommended a functional validation of the identified genes. You shuld also minimize the use of abbreviations,and the 6 genes should be spelled out in the text, and make more significance discussion.

Reviewer 1 ·

Basic reporting

Overall experimental designs are too easy to make a conclusion. Just gene sequence screening based on computer simulation is not enough to draw a conclusion. It it necessary to do more in-depth bench research.

Experimental design

We would expect more deep experimental design for the research in addition to in-silico experiments.

Validity of the findings

Current findings are not sufficient to support the conclusion mentioned in the manuscript.

Additional comments

I would not accept this manuscript.

Reviewer 2 ·

Basic reporting

see below

Experimental design

see below

Validity of the findings

see below

Additional comments

The authors have analyzed datasets of primary and metastasis tissue of colon cancer patients and identified genes that are crucial to the metastasis and survival of colon cancer.
Major comments:
1. Although the findings of this study have implication for colon cancer diagnosis and therapy, it is lack of functional validation of the target genes. This is a major limitation of this study.
2. The authors should provide more detailed introduction and discussion on each of the four hub genes as well as the six genes for prediction of overall survival of colon cancer, such as biological function, its implication in cancer development, and literature report, etc.

Minor comments:
The language of this manuscript is poor to follow. Some typical language mistakes are shown below, as well as suggested changes.
1. Change the title to “comprehensive analysis of metastasis-related genes reveals a gene signature predicting the survival of colon cancer”
2. Line 25: change to “the genes that are altered during…”
3. Line 26: add “are” after “that”
4. Line 37: change “used” to “using”
5. Line 39: change “demonstrate” to “has demonstrated”
6. Line 40: change “involving” to “involved”
7. Line 257: change “screen” to “has screened”

Reviewer 3 ·

Basic reporting

.

Experimental design

.

Validity of the findings

.

Additional comments

Authors in this study has screened gene profiles involved in the metastasis of colon cancer and further identified four genes. They also identified and further validated six-gene signature that could be served as an indicator to prognosis colon cancer. Overall, this type of study will help to identify new therapeutic targets for the genes that are not yet proved to be associated in colon cancer metastasis. All in all, it is an interesting study, however, following concerns should be addressed by the authors before this work gets published.

1. The manuscript is carelessly prepared, with numerous spelling, grammar, and formatting errors. It is not suitable to be published at the current version. Below are few weak sentences I found:
Line-62: `need to further study`
Line-146: `that relevant with`
Line-257; 1a gene profiles`. Etc.

2. Please rewrite the abstract. The following sentence is confusing –
` A gene signature was developed using the genes related to the overall survival (OS) of colon cancer patient in TCGA dataset. The prognostic value of gene signature was confirmed in validation dataset with 390 colon cancer patients`.

---

## Round 0.2 · Minor Revisions

Using bioinformatics from the existing database is a good approach, especially related to clinical sciences.

The manuscript is almost ready to accept for publication but the journal checks indicated that there is some overlap of the most recent revision with some published sources. This must be addressed before publication.

I am attaching a copy of the iThenticate report for your reference.

Reviewer 2 ·

Basic reporting

see below

Experimental design

see below

Validity of the findings

see below

Additional comments

The authors have properly addressed all questions brought up in the first round of review. The manuscript is in good condition for publication.

Reviewer 3 ·

Basic reporting

Authors addressed all my concerns

Experimental design

Authors addressed all my concerns

Validity of the findings

Authors addressed all my concerns

Additional comments

Authors addressed all my concerns

---

## Round 0.3 · Minor Revisions

Thanks for revising the manuscript to meet the quality standard of PeerJ.

# Staff Note: A staff check has shown that the current version of your manuscript contains an unacceptable level of text re-use from other sources. We appreciate that the same methods are often reused and that there are only so many ways to describe a procedure. We are mostly concerned with any unacknowledged text overlap, as well as verbatim re-use, and these issues are probably best addressed by paraphrasing the sections concerned. Under separate cover, we will email you the report showing the problems. In addition, we note that your article currently lacks a Conclusions Section. Please address these issues and resubmit #

---

## Round 0.4 · Minor Revisions

Your manuscript has been improved; however, (1) many English errors or space problems have been identified. I have marked a few of them in filled yellow in your v3, which need to be corrected, and a careful proofreading is needed. (2) The use of the available GSE biosets to perform bioinformatics to search for biomarkers is of merit. However, in your future studies, you may try Partek Genomics software and Illumina Correlation Engine for biosets analysis, but also need biological or clinical verification to impact the field.

---

## Round 0.5 · accepted · Accept

Thanks for your contribution and reversions. The GSE database is an available resources to do bioinformatic analysis in combination with clinical experiences.